# Mesoporous Silica Nanoparticle-Based Drug Delivery Systems for the Treatment of Pancreatic Cancer: A Systematic Literature Overview

**DOI:** 10.3390/pharmaceutics14020390

**Published:** 2022-02-10

**Authors:** Etienne J. Slapak, Mouad el Mandili, Maarten F. Bijlsma, C. Arnold Spek

**Affiliations:** 1Center of Experimental and Molecular Medicine, Cancer Center Amsterdam, University of Amsterdam, Amsterdam UMC, 1105 AZ Amsterdam, The Netherlands; m.elmandili@amsterdamumc.nl (M.e.M.); c.a.spek@amsterdamumc.nl (C.A.S.); 2Laboratory for Experimental Oncology and Radiobiology, Cancer Center Amsterdam, University of Amsterdam, Amsterdam UMC, 1105 AZ Amsterdam, The Netherlands; m.f.bijlsma@amsterdamumc.nl; 3Oncode Institute, 1105 AZ Amsterdam, The Netherlands

**Keywords:** MSN, PDAC, targeted therapy, drug delivery, antitumor, modification

## Abstract

Pancreatic cancer is a devastating disease with the worst outcome of any human cancer. Despite significant improvements in cancer treatment in general, little progress has been made in pancreatic cancer (PDAC), resulting in an overall 5-year survival rate of less than 10%. This dismal prognosis can be attributed to the limited clinical efficacy of systemic chemotherapy due to its high toxicity and consequent dose reductions. Targeted delivery of chemotherapeutic drugs to PDAC cells without affecting healthy non-tumor cells will largely reduce collateral toxicity leading to reduced morbidity and an increased number of PDAC patients eligible for chemotherapy treatment. To achieve targeted delivery in PDAC, several strategies have been explored over the last years, and especially the use of mesoporous silica nanoparticles (MSNs) seem an attractive approach. MSNs show high biocompatibility, are relatively easy to surface modify, and the porous structure of MSNs enables high drug-loading capacity. In the current systematic review, we explore the suitability of MSN-based targeted therapies in the setting of PDAC. We provide an extensive overview of MSN-formulations employed in preclinical PDAC models and conclude that MSN-based tumor-targeting strategies may indeed hold therapeutic potential for PDAC, although true clinical translation has lagged behind.

## 1. Introduction

### 1.1. Pancreatic Cancer

Pancreatic ductal adenocarcinoma (PDAC), a neoplasm of the ductal cells in the exocrine pancreas, accounts for around 85% of pancreatic cancer diagnoses [1,2,3]. In 2020 the incidence rate (496,000 cases) for PDAC was almost equal to the number of deaths (446,000), making PDAC the seventh leading cause of cancer-related death in both sexes worldwide [4]. Median survival rates of PDAC are low at 11–15 months for resectable pancreatic cancer, 6–10 months for locally advanced cancer, and only 3–5 months for metastatic disease [5]. The average 5-year overall survival is 10% [6].

Treatment of PDAC depends on its disease stage and comprises surgical resection, radiation therapy, chemotherapy, and supportive care. Surgical resection is the only treatment with curative potential [7]. Based on disease stage, PDAC patients are divided into three groups; resectable/borderline resectable (10–20% of cases), non-resectable/locally advanced (around 30% of cases), and metastatic (around 60% of patients). Resectable/borderline resectable patients may receive neoadjuvant chemotherapy in combination with radiotherapy or adjuvant chemotherapy after surgical resection [8,9,10]. Gemcitabine monotherapy, which has been the golden standard for adjuvant treatment of the latter patients, has recently been replaced with fluorouracil, leucovorin, oxaliplatin, and irinotecan combination therapy (FOLFIRINOX) for patients with good post-operative performance status. The PRODIGE-24 trial showed significantly increased disease-free survival and increased median overall survival (OS) of around 20 months for FOLFIRINOX-treated patients [9]. In non-resectable/locally advanced diseases, nab-paclitaxel or FOLFIRINOX chemotherapy is the standard treatment option. Although a small percentage of patients do become eligible for surgery, most patients show limited response and remain ineligible for surgical resection [11].

Systemic chemotherapy is also the standard treatment option in metastatic disease. Gemcitabine monotherapy, the golden standard for many years, has been replaced by FOLFIRINOX as a first-line treatment option, based on improved progression-free survival (PFS) and OS reported in the ACCORD-11 phase III trial [12]. In addition to FOLFIRINOX, the MPACT phase III clinical trial also demonstrated improved PFS and OS of nab-paclitaxel/gemcitabine combination therapy compared to gemcitabine monotherapy in metastatic disease [13]. However, it is important to note that FOLFIRINOX is only recommended for patients with good performance status due to its significant treatment-associated toxicity [14]. In older patients, or those with a lower performance status, the administration of nab-paclitaxel/gemcitabine combination therapy is preferred over FOLFIRINOX due to its lower cytotoxicity profile. In patients with poor performance status, gemcitabine-based therapy remains the only treatment option available, but many patients refrain from treatment in this stage due to the limited benefit and high toxicity [11].

The limited effect of current treatment modalities on the survival in PDAC may be explained by several factors, including poor delivery of chemotherapeutic agents and high toxicity profiles of existing drugs [15]. Of note, PDAC is characterized by a desmoplastic reaction, and PDAC tissue frequently consists of over 80% non-tumor cells, and typically a minority of the tumor mass is made up of tumor cells [16]. The physical barrier posed by the stroma results in poor delivery of chemotherapeutic agents to the tumor cells, thereby severely hampering treatment efficacy [17]. As a consequence of poor drug delivery, patients must receive high drug doses to reach effective levels in the tumor, but the efficacy of such treatments is hampered by systemic toxicity with subsequent dose limitations and early cessation of therapy. Indeed, of the patients receiving gemcitabine or nab-paclitaxel/gemcitabine combination therapy, around 60% and 70%, respectively, have to discontinue treatment [18]. Of note, treatment-associated toxicities result in supportive care costs that surpass the cost of first-line treatment in FOLFIRINOX and nab-paclitaxel/gemcitabine combination therapy [18].

### 1.2. Targeted Delivery

To prevent toxicity-dependent dose-limitations, targeted delivery of chemotherapeutic drugs to cancer cells without affecting healthy non-tumor cells is an attractive therapeutic avenue to pursue. Such an approach would not only largely reduce morbidity but may also increase the number of patients eligible for chemotherapy treatment and increase efficacy by boosting local drug concentrations in the tumor. Proof of concept for targeted therapy was obtained by coupling paclitaxel to albumin nanoparticles (nab-paclitaxel), which increased the intratumoral activity of paclitaxel compared to free paclitaxel in preclinical models [19]. After a Phase III trial in which nab-paclitaxel (with gemcitabine) was associated with significantly better survival rates than gemcitabine alone, nab-paclitaxel was approved by the FDA for the treatment of PDAC [13]. Targeting nab-paclitaxel to the tumor site was hypothesized to depend on the binding of the albumin-moieties to the protein Secreted Protein Acidic and Rich in Cysteine (SPARC/osteonectin/BM40) overexpressed by fibroblasts in the stromal compartment [20]. However, subsequent preclinical work showed that nab-paclitaxel delivery and antitumor activity is independent of SPARC [21,22], implying that the increased efficacy of nab-paclitaxel hinges on improved bioavailability rather than specific targeting. Nevertheless, as already outlined above, nab-paclitaxel remains widely used as the first-line treatment in the setting of PDAC due to its relatively favorable toxicity profiles.

Several alternative targeted delivery strategies have been explored in PDAC over the last decades. These strategies employ—amongst others—liposomes [23,24,25,26], poly(lactic-co-glycolic acid) (PLGA)-based polymeric nanoparticles [27,28,29], solid lipid nanoparticles [30,31,32], and mesoporous silica nanoparticles (MSNs) of which especially the use of MSNs seems an attractive approach. MSNs are nanoscale silica-based particles with a porous structure as implied by their name. This porous structure enables high drug-loading capacity and time-dependent drug release. Additional advantages of MSNs include the tunable particle and pore sizes, high biocompatibility, the possibility of functionalizing the inner core and outer surface, and the possibility of controlled release through the use of a *gatekeeper* system [33,34]. The use of a gatekeeper system allows the targeted delivery and spatial- and temporal release of, for instance, chemotherapeutics or RNAi (siRNAs or shRNAs) from MSNs specifically (in)to PDAC cells and can be achieved by internal and external stimuli, such as pH gradients, enzymes, light or magnetic field [34,35,36]. In this review, we explore the suitability of MSN-based targeted therapies in the setting of PDAC by providing a systematic literature overview.

## 2. Materials and Methods

To explore the potential promise of MSNs for the management of PDAC, a systematic literature search was performed in MEDLINE/PubMed, Web of Science Core Collection, and the EMBASE database. The combination of search terms ‘PDAC’, ‘pancreatic’ or ‘pancreatic ductal adenocarcinoma’ with ‘MSN’, ‘mesoporous’, or ‘silica’ was used to retrieve all papers that focus on MSNs in PDAC until November 30th, 2021. The exact search can be found below. All retrieved papers were screened by title and abstract for eligibility. Review papers, conference manuscripts, papers without full-text, papers not written in English or of poor quality, and papers that did not focus on MSNs or PDAC cytotoxicity were excluded. PubMed search query: (“pancreatic cancer” AND MSN) OR (“pancreatic cancer” AND mesoporous) OR (“pancreatic cancer” AND silica) OR (PDAC AND MSN) OR (PDAC AND Mesoporous) OR (PDAC AND silica) OR (pancreatic ductal adenocarcinoma AND silica) OR (pancreatic ductal adenocarcinoma AND mesoporous) OR (pancreatic ductal adenocarcinoma AND MSN)Web of Science Core Collection search query: ((((((((ALL = (“pancreatic cancer” AND silica)) OR ALL = (“pancreatic cancer” AND mesoporous)) OR ALL = (“pancreatic cancer” AND MSN)) OR ALL = (PDAC AND silica)) OR ALL = (PDAC AND mesoporous)) OR ALL = (PDAC AND MSN)) OR ALL = (“pancreatic ductal adenocarcinoma” AND silica)) OR ALL = (“pancreatic ductal adenocarcinoma” AND mesoporous)) OR ALL = (“pancreatic ductal adenocarcinoma” AND MSN)EMBASE database search query: pancreatic cancer’ AND msn OR (‘pancreatic cancer’ AND mesoporous) OR (‘pancreatic cancer’ AND silica) OR (‘pdac’ AND msn) OR (‘pdac’ AND mesoporous) OR (‘pdac’ AND silica) OR (‘pancreatic ductal adenocarcinoma’ AND msn) OR (‘pancreatic ductal adenocarcinoma’ AND silica) OR (‘pancreatic ductal adenocarcinoma’ AND mesoporous).

## 3. Results

A total of 457 papers were retrieved from the different databases (Figure 1). After removing duplicates, 140 eligible papers were identified that were thoroughly screened for experimental data on MSNs in PDAC. This resulted in the inclusion of 42 papers in this systematic review (Table 1). As outlined below, MSN-based targeted therapies may use classical MSNs or may exploit hybrid MSN-based strategies comprised of MSNs combined with a liposomal, gold, or magnetic iron oxide component. Both these systems are mainly used for cytotoxicity experiments but are also under consideration for imaging purposes. A schematic representation of the MSN-based nanoparticles and their applications can be seen in Figure 2.

### 3.1. Cytotoxicity of Classical MSNs in PDAC

The last decade has seen a rise in the number of studies exploring MSNs in PDAC [37,38,39,40,41,42,43,44,45,46,47,48,49,50,51,52,53,54,55,56,57]. Building on experience with MSNs in other tumor types [79,80,81], MSNs are preferentially surface modified to increase biodistribution and/or tumor uptake. Pioneering studies showed that surface modification by conjugation with polyethyleneimine (PEI) [37], folic acid [82], or monoclonal antibodies targeting anti-claudin4 and anti-mesothelin [83] indeed improved nanoparticle uptake by PDAC cells, whereas modification with polyethylene glycol (PEG) was shown to enhance biodistribution and circulation time in experimental animal models [40,42]. More recent studies use alternative surface modifications to target chemotherapeutics to PDAC tumors, and MSNs have been conjugated with transferrin [46,47,83], urokinase plasminogen activator [54], anti-GPC1, anti-tMUC1 [48], or V7 [84] peptides for this purpose. As envisioned, cellular uptake was increased using tumor-targeting moieties compared to controls lacking a modification both in vitro [46,47,83] and in vivo [48,54,84]. A recent study confirmed the importance of tumor-targeting surface modifications [84]. Utilizing V7 peptide-conjugated MSNs in an orthotopic PDAC model, MacCuaig and colleagues showed that active targeting of MSNs (i.e., including surface modifications to increase tumor cell uptake) outperforms passive targeting (i.e., no tumor targeting modifications on the MSNs) irrespective of nanoparticle size [84]. However, it is important to note that improved uptake and cytotoxicity in vitro does not always translate to similar findings in vivo, as PEGylated MSNs showed higher tumor uptake compared to PEG-transferrin-modified MSNs in one study [47]. More importantly, drug-loaded anti-tMUC1-conjugated MSNs outperformed MSNs lacking the anti-tMUC1 moiety in reducing tumor volume and weight in a syngeneic mouse model in which human tMUC expressing PDAC cells were implanted [48].

In addition to targeting MSNs to tumor cells, MSNs may also be surface modified to only release their cargo in the proximity of tumor cells. To this end, several so-called gatekeeper systems have been developed that prevent drug release in the circulation and/or at healthy, non-tumor-bearing organ sites. The addition of pH-sensitive gatekeepers such as chitosan, disulfide bonds, and poly(d,l-lactide-co-glycolide) showed pH-specific cargo release in vitro [45,46] and in tissue-mimicking phantoms [54,84]. The reduction in tumor weight and volume upon administration to PDAC bearing mice suggests that pH-based gatekeepers also hold promise for in vivo settings [43]. Unfortunately, no in vitro or in vivo experiments have been performed to compare the cytotoxicity of MSNs with and without a pH-sensitive gatekeeper to prove its superior sensitivity for the tumor microenvironment compared to healthy tissue. MSNs may also achieve specific drug release in the vicinity of tumor cells capped and locked by protease linkers that are specifically cleaved by tumor-enriched proteases. Indeed, conjugating MSNs with an ADAM9-responsive peptide linker more efficiently killed PDAC cells than white blood cells in vitro [50].

As opposed to tumor intrinsic properties, external cues can also be applied to remove the gatekeeper from MSNs thereby inducing drug release. Removing a thermo-responsive gatekeeper using an alternating magnetic field (AMF) led to rapid drug release and efficient PDAC cell death whereas no cell death was observed in the absence of AMF [41]. Alternatively, external stimuli may be applied to activate MSNs to induce cell death. Photonic stimulation of MSNs loaded with the photosensitizer ZnPcOBP caused a high phototoxic effect compared to free ZnPcOBP on PDAC cells in vitro. This effect was further enhanced by surface modification with Cetuximab, a monoclonal antibody that targets the Epidermal Growth Factor Receptor [44]. Of note, the observed photokilling of 

ZnPcOBP-loaded Cetuximab-conjugated MSNs correlated with (epidermal growth factor receptor (EGFR) expression levels in the PDAC cells. Similarly, the delivery of oxygen and the sonosensitizer IR780 by MSNs to the hypoxic tumor environment reduced tumor volume and improved survival in experimental animals upon sonodynamic therapy [38]. This method, however, relies on ultrasound irradiation and the presence of EGFR on tumor cells, which might not be translatable to real-world clinical routines and possibilities. Despite the substantial number of in vitro studies with MSNs in PDAC, only several papers study the potential of MSNs in preclinical animal models. To target, the stromal compartment, MSNs coated with PEI/PEG/LY364947 (a small molecule TGF-β inhibitor) were administered to tumor-bearing mice [40]. Of note, the number of pericytes in the stroma surrounding the tumor cells was reduced and subsequent treatment with PEGylated gemcitabine-loaded liposomes efficiently reduced tumor weight as compared to control mice that were only treated with gemcitabine-loaded liposomes [40]. Importantly, combination therapy of PEI/PEG/LY364947 coated MSNs with gemcitabine-loaded liposomes did not induce cytotoxicity (i.e., body weight loss or nephrotoxicity) as opposed to free gemcitabine. Of note, this promising study was already published in 2013, and no follow-up has yet been reported. Gao and co-workers employed MSNs loaded with the antifibrotic drug pirfenidone that were subsequently capped with gemcitabine to simultaneously target the stromal and tumor compartment combined with ultrasound destruction [43]. This intriguing approach almost completely halted tumor growth for three weeks and prolonged survival compared to both free gemcitabine and pirfenidone [43]. However, half of the mice succumbed to the disease after seven weeks, indicating that the observed tumor growth inhibition was an early response not sustained over time. In addition, although the increased cytotoxicity of the MSNs was not accompanied by any toxicity after three weeks of treatment, ultrasound destruction is well-known to induce damage to healthy tissues, limiting the applicability in clinical trials. One possibility to circumvent this caveat would be to monitor biological tissue damage using, for example, the IMWPE-PNN method [as described in Bei Liu et al. [85]]. The combination of cisplatin and gemcitabine is associated with high toxicity, yet recent clinical trials imply an added benefit of including cisplatin in existing PDAC treatment regimens [86,87]. Based on this notion, a very recent study designed MSNs with cisplatin and gemcitabine prodrugs to the inner and outer surface, respectively. Systemic administration of these MSNs to two genetic tumor-bearing mouse models significantly suppressed tumor growth and eliminated the off-target toxicities of the highly toxic chemotherapy combination. By mimicking advanced stages of PDAC in vivo over a study course of three months, they were able to show therapeutic effect by a decrease in pancreas weight, attributed by a reduction of the tumor mass. [48]. In vivo study designs like these might improve the clinical translation.

In addition to conventional chemotherapeutics, MSNs also open up new avenues for drugs whose clinical potential is hampered by their hydrophobicity and consequent bio-distribution. The clinical efficacy of curcumin, a candidate anticancer drug [88] that potentiates the effect of gemcitabine [89,90], is limited by its poor solubility. Loading curcumin into MSNs was found to inhibit tumor growth and minimize distant metastasis in a subcutaneous xenograft model [47]. Of note, subsequent administration of gemcitabine potentiates the effect of curcumin-loaded MSNs in vitro, but in vivo validation is lacking [47].

Overall, a picture emerges in which classical MSNs are attractive vehicles to deliver drugs to PDAC tumors. Different surface modifications have shown promising characteristics in preclinical PDAC models, and several MSN formulations warrant follow-up in future clinical studies.

### 3.2. Liposome-Coated MSNs

Amongst all nanomedicine platforms, liposomes—spherical vesicles composed of a lipid bilayer—are most used and several FDA-approved liposome formulations (most notably liposomal irinotecan, Onivyde, in the setting of PDAC) are used in the clinic. Based on the favorable characteristics of liposomes, several papers describe the coating of MSNs with a lipid bilayer to improve stability after systemic administration, thereby overcoming one of the major limitations of MSNs in vivo. The majority of lipid membrane-enhanced MSNs lack a targeted delivery moiety [58,59,60,61,63,64,65,66,68], however iRGD- [62] and cyclosporine A-conjugated [67] liposome-coated MSNs have been designed to improve tumor targeting and cellular uptake. Non-targeted irinotecan-loaded liposome-coated MSNs consistently outperform irinotecan-loaded liposomes, including FDA-approved Onivyde, in terms of drug delivery, cytotoxicity, survival as well by reducing bone marrow, gastrointestinal and liver toxicity [64,66]. Indeed, compared to free drug and Onyvide, the liposome-coated MSNs amounted to a 79- and 8.7-fold increase in tumor drug content, respectively. In line, irinotecan-loaded liposomes significantly increased survival compared to Onyvide in an orthotopic PDAC model [66]. Modifying the liposome-coated MSNs with a tumor-homing and penetrating iRGD-peptide enhanced survival even further and resulted in reduced metastasis [62]. The significant improvement of irinotecan-loaded liposome-coated MSNs over the last five years resulting in their superiority over Onyvide, poses it as an interesting candidate for progressing to clinical testing. Further research showed that combining different drugs in lipid-modified MSNs greatly improves tumor reduction compared to free drugs or corresponding monotherapies. Indeed, gemcitabine/paclitaxel-loaded MSNs outperform gemcitabine-loaded MSN monotherapy and combination therapy of free gemcitabine and nab-paclitaxel [58]. Moreover, co-administration of palbociclib- and hydroxychloroquine-loaded MSNs [59], or indoximod- and oxaliplatin-loaded MSNs [68] reduced PDAC growth more efficiently compared to mono MSN therapy or free drug combinations. Besides, the co-delivery of chemotherapeutic-loaded liposome-coated MSNs can also be adjusted to facilitate photothermal and photodynamic-induced cancer cell apoptosis [67]. The increased uptake dependent on cyclosporine A conjugation improved the apoptotic effects of bortezomib in combination with the cytotoxic effects of the near infrared (NIR) dye IR-820 upon NIR irradiation in a subcutaneous PDAC model [67]. In conclusion, liposome-coated MSNs confer great versatility and show great promise in preclinical research, notably by outperforming the FDA-approved classical liposomal formulation Onivyde upon loading with irinotecan.

### 3.3. Gold-MSN Hybrid Nanocarriers

Gold-MSN hybrid nanocarriers are typically employed for imaging purposes (see below for details), but they may also be used to potentiate treatment response. By extending the lifetime of highly toxic singlet oxygen species necessary for photosensitization, the cytotoxic potential of the involved photosensitizer molecules is increased [91]. Indeed, the conjugation of gold-nanoparticles to MSNs loaded with the photosensitizer methylene blue decreased PANC-1 cell viability following photodynamic therapy (PDT) compared to MB-loaded MSNs lacking a gold nanoparticle tethered to the outer layer in vitro [71]. The superior efficacy of gold-MSNs has been confirmed in preclinical animal models by two research groups [69,70]. Both studies employing gold-MSNs in vivo used conjugated MSNs, with IGF1 [69] or transferrin [70], to increase their cellular uptake. In a patient-derived xenograft PDAC mouse model, gemcitabine-loaded IGF1-conjugated gold-MSNs reduced tumor growth by around 70% [69]. Combining the gold-MSNs with photothermal therapy further enhanced efficacy leading to complete eradication of the xenograft and an astounding survival rate of 100% [69]. Next to their remarkable antitumor efficacy, the gold-MSNs did not seem to induce any cytotoxicity off-target. Albeit promising, intratumoral injection for photoablation limits the therapeutic efficacy to the primary tumor, leaving metastatic foci unharmed. Moreover, such a treatment would be hard to implement in the clinic and would require incorporation in local ablation modalities. In line with these intriguing data, gemcitabine-loaded transferrin-conjugated gold-MNs also greatly enhanced chemosensitivity of PDAC cells and induced effective regression of human pancreatic cancer xenografts in mice by the combination of photothermal- and chemotherapy [70]. The impressive antitumor efficacy was in part attributed to an increased penetration of gemcitabine after photothermal therapy. However, it is important to note that photothermal ablation is not readily translatable to human PDAC due to the tissue absorption of laser light causing a decrease in intensity of approximately 10-fold every 2 cm deeper [92]. Consequently, it would be necessary to address whether MSN-based therapies involving photothermal ablation by laser light can be applied in a clinical setting. Additionally, it would be interesting to assess if loading gold-MSNs with drugs with a higher cytotoxic activity towards PDAC cells, such as nab-paclitaxel, may even further reduce tumor growth.

### 3.4. Magnetic Iron Oxide-MSN Hybrid Nanocarriers

A relatively recently developed hybrid MSN nanocarrier system combined MSNs with an iron oxide component [73,74,75,76,77,78]. These hybrid MSNs allow simultaneous MRI contrast imaging and drug delivery, thereby enabling the visualization of therapy efficacy in a non-invasive manner. Although the number of papers describing theranostic magnetic iron oxide-MSNs is limited, they seem to hold promise in the setting of PDAC. Indeed, tumor microenvironment-triggered release of poorly water-soluble camptothecin molecules from magnetic iron oxide-MSNs reduced tumor growth in vivo [73]. This study, however, was limited to 13 days, and, therefore, long-term efficacy must be demonstrated in future research. The safety and biocompatible nature of the magnetic iron oxide-MSNs was confirmed by histological analysis, and no overt signs of toxicity were observed in other organs. As described above, in PDAC, MSNs may be employed in a ‘two-hit’ approach in which the first hit targets the stroma to then improve tumor delivery of drugs carried by the MSNs (second hit). Based upon this notion, Li and colleagues treated tumor-bearing mice with magnetic iron oxide-MSNs loaded with losartan that inhibits type I collagen and hyaluronic acid present in PDAC stroma. Mono treatment of losartan-loaded magnetic iron oxide-MSNs marginally reduced tumor volume, but subsequent treatment with gemcitabine-loaded magnetic Iron Oxide-MSNs very efficiently diminished tumor growth by over 70%. Notably, monotherapy with gemcitabine-loaded magnetic iron oxide-MSNs was less effective and reduced tumor growth by around 40% [75]. Unfortunately, the endpoint was after only three weeks, limiting the observation of long-term effects. Similar to surface modifications described above, magnetic iron oxide hybrid MSNs may also be further modified to increase specificity or efficacy. As the first example of such an approach, Sun and colleagues showed that adding a c(RGDfE) moiety improved the cellular uptake by PDAC cells [77]. Whether this modification or alternative modifications used for targeting purposes in classical MSNs, also improves efficacy in preclinical PDAC animal models need to be established. Preclinical assessment needs to be improved by extending the treatment and follow-up period of in vivo experiments. Furthermore, the wide variety of magnetic iron oxide molecules used in hybrid-MSNs complicate the ability to compare studies and address superiority, future standardization experiments might be particularly useful for this type of MSNs.

## 4. Conclusions

The preclinical studies discussed in this systematic review suggest that MSN-based tumor-targeting strategies may hold therapeutic potential for PDAC. Indeed, MSNs-based therapies show antitumor activity in PDAC mouse models and seem to reduce adverse toxicity. Several issues need to be kept in mind before MSNs can move forward to clinical development in PDAC management. The MSNs employed in the (preclinical) studies are rather variable with respect to their synthesis and surface modifications, and no direct comparisons have been made between these MSNs. Indeed, the MSN formulations have been tested in different preclinical models with varying drug concentrations, controls, endpoints, and treatment modalities. Hence, it will be pivotal to compare different MSN formulations head-to-head in similar models with predefined endpoints. Only when such studies have been performed will we be able to select the most promising MSN-based strategy to test in clinical studies. Unfortunately, clinical translation remains slow. Even though the safety of MSNs has been widely demonstrated in clinical trials, it has taken over more than two decades for gold-MSNs to reach clinical trials [93]. This slow progression may be explained by the over-interpretation of results combined with the majority of papers not passing the critical assessment of their translatable applicability. Additionally, multiple MSN-based nanoplatforms showing pre-clinical promise are not followed up or improved over time, raising the question of whether follow-up was not performed or whether it yielded less than encouraging results. Another important limitation of several of the discussed studies is the lack of proper controls to address potential side effects of the MSN formulations. To conclude that MSN-loaded drugs confer reduced cytotoxicity, it is pivotal to include relevant control cells in vitro and proper toxicity readouts in vivo. As most chemotherapeutics show bone marrow toxicity induces neuropathy and diarrhea, preclinical mouse models should be designed to assess these common side effects. In vitro, one should consider including blood, neuronal, or (gut) epithelial cells to assess the effect of the MSNs on the relevant cell types. Irrespective of these latter considerations, MSN-based targeted therapies seem to hold promise for treating PDAC, a disease that is in dire need of improved therapeutic options.

## Figures and Tables

**Figure 1 pharmaceutics-14-00390-f001:**
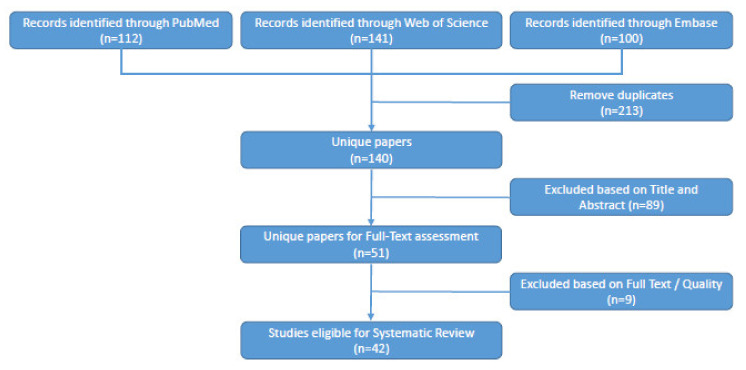
Flowchart explaining the systematic literature search. All retrieved papers were screened, and duplicates were removed, followed by exclusion based on title and abstract, full text, or quality.

**Figure 2 pharmaceutics-14-00390-f002:**
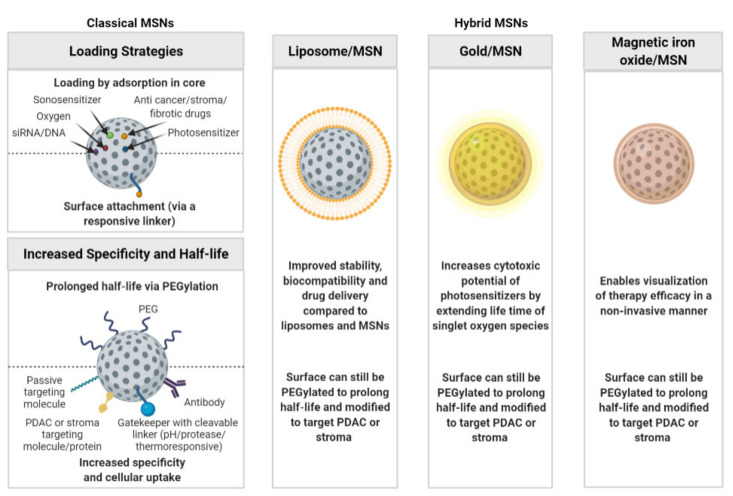
Diagram summarizing loading strategies of different molecules (siRNA/DNA, oxygen, sonosensitizer (e.g., IR780), anticancer drugs (e.g., gemcitabine, cisplatin, curcumin, irinotecan, paclitaxel, palbociclib, and oxaliplatin), anti-stroma drugs (e.g., TGF-β inhibitor), antifibrotic drugs (e.g., pirfenidone), photosensitizers (e.g., ZnPcOBP and methylene blue), methods to prolong half-life, increase specificity and cellular uptake (passive targeting molecules (e.g., polyethyleneimine), PDAC targeting molecules/proteins (e.g., folic acid, transferrin, urokinase plasminogen activator, V7 peptides, cyclosporine A, IGF1, c(RGDfE), and CCKBR aptamer), stroma targeting molecules (e.g., iRGD), gatekeeper with pH sensitive linkers (e.g., chitosan, disulfide bonds, and poly(D,L-lactide-co-glycolide) (PLGA)), gatekeepers with protease linkers (e.g., ADAM9-responsive linker capped with avidin, thermoresponsive gatekeeper (e.g., aliphatic azo group capped with β-cyclodextrin), antibodies (e.g., tMUC-antibody, GPC1-antibody, Cetuximab, anti-claudin 4, and anti-mesothelin), and hybrid MSNs.

**Table 1 pharmaceutics-14-00390-t001:** MSN-based Therapies for Improved Drug Delivery in PDAC.

MSN	Modification	Aim of Modification	Experimental Model	Drug/ Treatment	Main Outcome	Ref.
MSN	PEG PEI	↑ Biodistribution ↑ Uptake	• PDAC cells	Paclitaxel	•↑ cellular uptake compared to unmodified MSN •↑ cytotoxicity compared to free drug	[37]
MSN	FC-Chain	Oxygen Delivery	• PDAC cells • Subc. Mouse	Sonodynamic Therapy	•↓ cell proliferation of multi-treatment MSNs compared to single treatment MSNs •↓ tumor volume compared to untreated, single-treatment MSNs •↑ improved survival compared to untreated, single-treatment MSNs	[38] *
MSN	Folate	↑ Uptake	• Subc. mouse	Camptothecin	•↓ tumor volume compared to untreated, free drug and unmodified MSNs	[39]
MSN	LY364947 PEG PEI	TGF-β inhibition ↑ Biodistribution ↑ Uptake	• PDAC cells • Orth.. mouse • Subc. mouse	LY364947 (TGFb inhibitor)	•↓ pericytes coverage compared to free TGF-β inhibitor •↑ delivery Gemcitabine-loaded-liposomes compared to single-treatment •↓ tumor volume compared to untreated, free drug and unmodified MSNs	[40] *
MSN	ACVA β-cyclodextrin	Cargo Release Gatekeeper	• PDAC cells	Doxycycline	• Thermoresponsive release of cargo •↑ cytotoxicity compared to untreated and empty MSN	[41]
MSN	PEG	↑ Biodistribution	• PDAC cells	Curcumin	•↑ curcumin formulation	[42]
MSN	Gemcitabine	Gatekeeper	• PDAC cells • Subc. Mouse	Pirfenidone/Gemcitabine	•↓ expression stromal components •↓ IC50 compared to free drug •↓ tumor volume compared to untreated and free drug •↑ survival compared to untreated and free drug • No adverse effects major organs after three weeks of treatment	[43] *
MSN	Cetuximab Imidazole PEG	↑ Uptake Gatekeeper ↑ Biodistribution	• PDAC cells	ZnPcOBP (Photodynamic Therapy)	•↑ cellular uptake compared to unmodified MSN •↑ cytotoxicity compared to empty and unmodified MSN	[44] *
MSN	Chitosan	Cargo Release	• PDAC cells	N6L (Nuceolin antagonist)	• pH-sensitive cargo release	[45]
MSN	Transferrin Chitosan PLGA	↑ Uptake Cargo Release Cargo Release	• PDAC cells	Gemcitabine	•↑ cytotoxicity compared to unmodified MSN • pH-sensitive cargo release	[46]
MSN	Transferrin PEG	↑ Uptake ↑ Biodistribution	• PDAC cells • Subc. mouse	Curcumin	•↑ cellular uptake compared to unmodified MSN •↓ tumor growth and metastasis compared to free drug and unmodified MSN	[47]
MSN	tMUC-antibody PEG PEI	↑ Uptake ↑ Biodistribution ↑ Uptake	• PDAC cells • Genetic mouse	Gemcitabine-/ cisplatin prodrug	•↑ cellular uptake compared to unmodified MSN •↑ cytotoxicity double-loaded MSNs compared to a single drug and mixed •↑ cellular uptake compared to unmodified MSN •↓ tumor volume and weight compared to control, free drug and unmodified MSN • No adverse effects major organs	[48] *
MSN	Cetuximab PEG	↑ Uptake ↑ Biodistribution	• PDAC cells • Orth. mouse	Zinc phthalocyanine	•↑ cellular uptake compared to free drug and unmodified MSN •↓ tumor volume	[49]
MSN	ADAM9-linker Biotin-avidin	Cargo Release Gatekeeper	• PDAC and white blood cells	Paclitaxel	•↑ cytotoxicity in PDAC compared to white blood cells	[50]
MSN			• PDAC cells • Intraperi. mouse	Paclitaxel	•↑ cellular drug concentration compared to free drug •↑ drug concentration in tumor compared to free drug	[51]
MSN	L-arginine	CO_2_ adsorption/ release	• PDAC cells • Subc. mouse	Sonodynamic Therapy	•↑ cytotoxicity compared to single-treatment •↓ tumor volume compared to single-treatment	[52]
MSN	GPC1-antibody	↑ Uptake	• PDAC cells	Gemcitabine/ Ferulic Acid	•↑ cytotoxicity compared to unmodified MSNs	[53]
MSN	Chitosan UPA	Cargo Release Cargo Release	• PDAC cells	Gemcitabine	• pH-specific cargo release	[54]
MSN			• PDAC cells	Doxorubicin	• Delivery of doxorubicin to cytoplasm PDAC cells	[55]
MSN	Quantum Dots	Cargo Loading	• PDAC cells	Doxorubicin/ Camptothecin	•↑ cytotoxicity multidrug-loaded MSNs compared to single drug-loaded MSNs	[56]
MSN			• PDAC cells	Paclitaxel	• Dose-dependent cytotoxicity	[57]
Lipo-MSN			• PDAC cells • Subc. mouse • Orth. mouse	Paclitaxel/ Gemcitabine	• Synergy of paclitaxel and gemcitabine upon co-delivery •↑ tumor shrinkage compared to free drug, MSN-loaded and combination therapy •↓ primary tumor growth and metastasis	[58]
Lipo-MSN	PEG	↑ Biodistribution	• PDAC cells • Subc. mouse • Orth. mouse	Palbociclib/ Hydroxy-chloroquine	•↑ cytotoxicity co-delivery compared to free and single drug MSNs •↓ tumor size co-delivery compared to free drug and single drug MSNs •↓ tumor size co-delivery compared to free drug and single drug MSNs	[59]
Lipo-MSN			• Orth. mouse	Irinotecan	•↓ tumor size and improved survival compared to free drug and Onivyde •↓ liver, GIT, and bone marrow toxicity	[60]
Lipo-MSN			• PDAC cells	P1A1 (Platinum-acridine)	•↑ cytotoxicity compared to empty MSN and free drug	[61]
Lipo-MSN	iRGD PEG	↑ Uptake ↑ Biodistribution	• Orth. mouse	Irinotecan	•↑ cellular uptake compared to unmodified Lipo-MSN •↑ survival and ↓ metastasis compared to unmodified Lipo-MSN	[62]
Lipo-MSN	PEG	↑ Biodistribution	• Orth. mouse	Oxaliplatin/ fDACHPt	•↓ tumor weight and metastasis, improved survival compared to free drug	[63]
Lipo-MSN	PEG	↑ Biodistribution	• Orth. mouse	Irinotecan	•↓ tumor weight and metastasis compared to free drug •↑ survival compared to free drug and Onivyde	[64] *
Lipo-MSN	PEG	↑ Biodistribution	• Orth. mouse	Irinotecan	•↓ tumor weight and metastasis, improved survival compared to free drug •↓ liver, GIT, and bone marrow toxicity	[65]
Lipo-MSN	PEG	↑ Biodistribution	• Orth. mouse	Irinotecan	•↓ tumor weight and metastasis compared to free drug and Onivyde	[66] *
Lipo-MSN	Cyclosporine A PEG	↑ Uptake ↑ Biodistribution	• PDAC cells • Subc. mouse	Bortezomib/ IR-820 (Photothermal Therapy)	•↑ cellular uptake compared to unmodified Lipo-MSN •↓ tumor volume and growth compared to free drug and unmodified Lipo-MSN	[67]
Lipo-MSN	PEG	↑ Biodistribution	• Subc. mouse • Orth. mouse	Oxaliplatin/ Indoximod	•↓ tumor size and metastasis compared to free drug, single-drug Lipo-MSNs •↑ survival compared to free drug, single-drug Lipo-MSNs •↓ tumor size and metastasis compared to free drug, single-drug Lipo-MSNs •↑ survival compared to free drug, single-drug Lipo-MSNs	[68]
Gold-MSN	IGF-1	↑ Uptake	• PDAC cells • Subc. mouse	Gemcitabine/Perfluorohexane	•↑ cytotoxicity compared to untreated, free drug, unmodified MSNs • Complete response compared to untreated, free drug, unmodified MSNs • No adverse effects major organs	[69] *
Gold-MSN	Transferrin PEG	↑ Uptake ↑ Biodistribution	• PDAC cells • Subc. mouse	Gemcitabine	•↑ cellular uptake compared to unmodified Gold-MSN •↓ tumor volume compared to free drug, empty and unmodified Gold-MSN	[70] *
Gold-MSN			• PDAC cells	Methylene Blue (Photodynamic Therapy)	•↑ cytotoxicity Gold-modified MSNs compared to unmodified	[71]
Gold-MSN	V7-peptide Chitosan	↑ Uptake Cargo Release	• PDAC cells	Gemcitabine	•↑ cytotoxicity compared to free drug and empty MSNs	[72]
Iron-MSN			• PDAC cells • Orth. mouse	Camptothecin	•↑ cytotoxicity compared to free drug, empty MSN, and unmodified MSN •↓ tumor volume compared to untreated and free drug • No adverse effects major organs	[73] *
Iron-MSN	Dicarboxylic acid	Cargo Release	• PDAC cells	Cisplatin	•↑ cytotoxicity compared to free drug •↓ cytotoxicity nonmalignant human pancreatic duct cells	[74]
Iron-MSN			• PDAC cells • Subc. Mouse	Gemcitabine/ Losartan	•↑ cytotoxicity compared to free drug •↓ tumor weight and volume compared to monotherapy • No adverse effects major organs	[75] *
Iron-MSN			• PDAC cells	Doxycycline	• Dose-dependent cytotoxicity	[76]
Iron-MSN	c(RGDfE) PEG	↑ Uptake ↑ Biodistribution	• PDAC cells	Gemcitabine	•↑ uptake compared to unmodified Iron-MSNs	[77]
Iron-MSN	CCKBR aptamer G16 PEG citrate	↑ Uptake ↑ Uptake ↑ Biodistribution ↑ Biodistribution	• PDAC cells • Orth. mouse	FdUMP/ dFdCMP	•↓ proliferation compared to free drug and empty MSNs •↓ thymidylate synthase levels compared to unmodified MSNs	[78]

• = in vitro, • = in vivo, Orth. = Orthotopic, Subc. = Subcutaneous, Intraperi. = Intrapertitoneal, ↑ = increased, ↓ = decreased, * indicates particularly relevant publication.

## Data Availability

Not applicable.

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
