# Peer review of "Mesoporous Silica Nanoparticle-Based Drug Delivery Systems for the Treatment of Pancreatic Cancer: A Systematic Literature Overview"

_pharmaceutics, 2022, doi:10.3390/pharmaceutics14020390_

Round 1

Reviewer 1 Report

The manuscript systematically covers the lieteraute on use of Mesorposurs nanoparticles based delivery systems for pancreatic cancer.

The caption for figure 1 must be detailed. 

Any specific time period was chosen for the selection of articles? Please specify the time or state if not considered. 

Reviewer 2 Report

The authors have carefully done a literature search using 3 databases of Mesoporous Silica Nanoparticles for drug delivery for PDACreatic cancer.  This search after culling resulted in only 45 papers.  All are preclinical. 

The clinical drug situation with PDAC is well-presented in the introduction.  The article is also comprehensive, with a table presentation, and spending time to describe outstanding results in the text and organizing the results into classical MSNs, liposome-coated MSNs, gold nanoparticle MSNs, and  iron oxide np MSNs. The study is very valuable since it saves time for researchers interested in this field of application to quickly gain a comprehensive view of the landscape.  This review is worthy of publication after the following points are addressed.

  1. PDAC is a difficult cancer to treat. Preclinical results with MSNs show some promise in general, but rarely pass critical assessment of their translatable ability.  This review is far too optimistic about the preclinical results.  For example the authors characterize one paper as “magnetic iron oxide-MSNs resulted in partial or even complete regression of the tumors in vivo [66]”  Upon examination of this reference, the study of tumor growth was limited to 13 days!  Hardly qualifying as “complete regression”.  These short term studies are fraught with tumor regrowth at later times.  Another example is the authors’ synopsis stating “Combining the gold-MSNs with phototermal therapy further enhanced efficacy leading to complete eradication of the xenograft and an astounding survival rate of 100% [62]”.  This treatment uses direct intratumoral injection of a gel, and photoablation with a  laser.  With photoablation it is relatively easy to ablate a tumor by cooking it.  However this is not readily translatable to human PDAC due to the tissue absorption of laser light.  Intensity falls off a factor of ~10 at 2 cm with 808nm light.  It also involves intratumoral injection which is not particularly suitable for metastasizing PDAC.  Another example, the authors describe one study [Ref 35] that it “almost completely halted tumor growth”.  Sounds very promising, and is a good advance, but in fact it relies on ultrasound destruction that is not mentioned.  Ultrasound tissue destruction has many problems of normal tissue damage.  Also, even though tumor growth was slowed for the 3 weeks only reported, half of the animals died by 7 weeks, indicating that this method had an early response, but certainly did not “almost completely halt tumor growth”.  While the authors spin studies to sound very promising, there are nearly in all cases serious problems.  The data is not critically presented and this must be fixed.
  2. Similarly, the very promising GEM/PTX LB-MSNP construct “effectively inhibited primary tumor growth and eliminated metastatic foci” in a preclinical PDAC-1 orthotopic model. This was published in 2015. Where is this now after 7 years?  Is it in the clinic or were there problems?  Also this paper does not report survival, only tumor size measurements to 38 days.  The authors should comment on the follow-up of promising results published.
  3. The table summarizing and classifying published studies is a great idea. However it needs a lot of work. It is difficult to compare results without reading each text statement and looking up the superscript meanings.  Important categories rather than columns are given as tiny hard to read superscripts.  I suggest more columns with check boxes for items like in vitro, in vivo, targeted, and column for drug(s) used.  Remove wording ‘increased’ ‘improved’ since all papers claim something like this.  Maybe add ‘unique or important result’.  Add column to list targeting agent.  Remove the category ‘miscellaneous (e)’ which is meaningless.
  4. While this article focuses on MSN approaches, competing nanoparticle constructs such as PLGA, liposomes, SLN, etc. are being pursued. Please add some additional reference to those fields. Gene delivery using MSNs should be mentioned.
  5. Please mention status of clinical trials of MSNs, e.g., Clinical translation of silica nanoparticles. Janjua TI, et al. Nat Rev Mater. 2021. PMID: 34642607.
  6. Please add an instructive diagram (drawing) summarizing MSNs for PDAC.

Reviewer 3 Report

This review reports in a systematic way several studies on the potential of silica mesoporous nanoparticles as drug carriers for chemotherapy of pancreatic cancer. The review is well written and explores in a very exhaustive way, supported by recent references, this interesting topic. The summarizing table is very useful. The nanoparticles modification for therapeutic strategies are also well described.  The paper should be published in the present form. 

Round 2

Reviewer 2 Report

The authors have adequately answered review issues and modified the manuscript accordingly.